# Bone Marrow Mast Cell Antibody-Targetable Cell Surface Protein Expression Profiles in Systemic Mastocytosis

**DOI:** 10.3390/ijms20030552

**Published:** 2019-01-28

**Authors:** Noelia Dasilva-Freire, Andrea Mayado, Cristina Teodosio, María Jara-Acevedo, Iván Álvarez-Twose, Almudena Matito, Laura Sánchez-Muñoz, Carolina Caldas, Ana Henriques, Javier I. Muñoz-González, Andrés C. García-Montero, J. Ignacio Sánchez-Gallego, Luis Escribano, Alberto Orfao

**Affiliations:** 1Cancer Research Centre (IBMCC, USAL-CSIC), Cytometry Service (NUCLEUS) and Department of Medicine, University of Salamanca, 37007 Salamanca, Spain; noeliadf@usal.es (N.D.-F.); amayado@usal.es (A.M.); jmunogon@usal.es (J.I.M.-G.); angarmon@usal.es (A.C.G.-M.); joseignaciosanchezgallego@gmail.com (J.I.S.-G.); escribanomoraluis@gmail.com (L.E.); 2Institute of Biomedical Research of Salamanca (IBSAL), 37007 Salamanca, Spain; mariajara@usal.es; 3Biomedical Research Networking Centre on Cancer–CIBER-CIBERONC, Institute of Health Carlos III, 28029 Madrid, Spain; 4Spanish Network on Mastocytosis (REMA), 45071 Toledo and 37007 Salamanca, Spain; ivana@sescam.jccm.es (I.Á.-T.); amatito@sescam.jccm.es (A.M.); lsmunoz@sescam.jccm.es (L.S.-M.); carolina.caldas@usal.es (C.C.); ana_fhen@hotmail.com (A.H.); 5Department of Immunohematology and Blood Transfusion, Leiden University Medical Center, 2333 Leiden, The Netherlands; c.i.teodosio@lumc.nl; 6Sequencing DNA Service (NUCLEUS), University of Salamanca, 37007 Salamanca, Spain; 7Institute for the Study of Mastocytosis of Castilla La Mancha (CLMast), Virgen del Valle Hospital, 45071 Toledo, Spain

**Keywords:** hematology, immunology, systemic mastocytosis, monoclonal antibodies, cell therapy and immunotherapy, antibody-targetable cell surface membrane proteins, immuno-phenotyping

## Abstract

Despite recent therapeutic advances, systemic mastocytosis (SM) remains an incurable disease due to limited complete remission (CR) rates even after novel therapies. To date, no study has evaluated the expression on SM bone marrow mast cells (BMMC) of large panel of cell surface suitable for antibody-targeted therapy. In this study, we analyzed the expression profile of six cell-surface proteins for which antibody-based therapies are available, on BMMC from 166 SM patients vs. 40 controls. Overall, variable patterns of expression for the markers evaluated were observed among SM BMMC. Thus, CD22, CD30, and CD123, while expressed on BMMC from patients within every subtype of SM, showed highly variable patterns with a significant fraction of negative cases among advanced SM (aggressive SM (ASM), ASM with an associated clonal non-MC lineage disease (ASM-AHN) and MC leukemia (MCL)), 36%, 46%, and 39%, respectively. In turn, CD25 and FcεRI were found to be expressed in most cases (89% and 92%) in virtually all BMMC (median: 92% and 95%) from both indolent and advanced SM, but with lower/absent levels in a significant fraction of MC leukemia (MCL) and both in MCL and well-differentiated SM (WDSM) patients, respectively. In contrast, CD33 was the only marker expressed on all BMMC from every SM patient. Thus, CD33 emerges as the best potentially targetable cell-surface membrane marker in SM, particularly in advanced SM.

## 1. Introduction

Systemic mastocytosis (SM) consists of a heterogeneous group of clonal mast cell (MC) disorders [1,2] that vary from indolent cases—e.g., indolent SM (ISM)—and that impair the quality of life of patients with a normal life-expectancy, to advanced forms of the disease—e.g., aggressive SM (ASM), SM with an associated clonal non-MC lineage disease (SM-AHN), and mast cell leukemia (MCL)—associated with a significantly shortened survival [3]. In the vast majority of SM patients (>90%), the clonal nature of pathological MC can be demonstrated by the presence of the stem cell factor receptor gene (*KIT)* D816V mutation [4], except for well-differentiated SM (WDSM) patients [5] and a fraction of MCL [6]. This mutation results in constitutive activation of *KIT*, which leads to aberrantly sustained proliferative and survival signals in neoplastic MC, that ultimately lead to their expansion and accumulation in distinct tissues, particularly in the skin and bone marrow (BM) [7].

For decades, conventional chemotherapeutic regimens administered to patients with advanced forms of SM have proven to be of limited benefit with low complete remission (CR) rates and only minor improvement on survival [7,8,9]. Wild-type *KIT* can be currently targeted by a progressively higher number of small tyrosine-kinase inhibitor (TKI) molecules including some that—e.g., midostaurin (PKC412) or imatinib—have proven beneficial for SM [10,11,12,13]. However, overall CR rates, even with these new drugs still remain low, except among the few WDSM patients presenting with mutations at exons 9 and 10 of *KIT* [14,15,16,17,18]. Altogether, this highlights the need for further improvement in the treatment of SM, particularly for advanced SM [19]. In recent years, immunotherapy, including immunotherapy strategies based on targeting cell-surface membrane proteins, has proven to be of great clinical benefit and has become a cornerstone in the treatment of an increasingly higher number of distinct hematologic malignancies [20]. However, their clinical use in SM remains very limited [21].

At present it is well-known that multiple factors are involved in determining the response to antibody-based therapies. Despite this, a pre-requisite to achieve an optimal response to such therapies is the expression of the targeted protein on the whole tumor MC population in a per patient basis [22,23]. Multiple studies have described the overall patterns of expression of many proteins on the surface membrane of both normal and SM MC, for which distinct therapeutic antibody-based molecules have been designed, evaluated, and approved for their use in tumoral and non-tumoral human diseases [22,24,25]. These antibody-targetable cell surface membrane proteins include CD22, CD25, CD30, CD33, CD123, and FcϵRI, which have all been found in tumor MC from SM patients [22] (Table 1). Some of these markers have even been targeted by therapeutic antibodies outside clinical trials, usually in small series of SM patients and single case reports, with variable responses [26,27,28,29]. However, these immuno-phenotypic studies failed to provide information about the patterns of expression of the involved markers within individual patients and across distinct disease subtypes, particularly among advanced SM cases.

In this study, we provide, for the first time, detailed information about the patterns of expression on BMMC from SM patients (*n* = 166) with distinct World Health Organization (WHO) diagnostic categories of the disease, of six surface proteins known to be expressed on BMMC, and for which the US Food and Drug Administration (FDA) and/or European Medicines Agency (EMA)-approved for safety antibody therapies are available for humans (CD22, CD25, CD30, CD33, CD123 and FcϵRI). Our major goal was to identify, among all the markers, those that would show the highest and broadest expression on BMMC from individual patients across the distinct variants of the disease, particularly in advanced SM, which makes them potentially suitable candidates for currently available antibody-targeted therapies, whenever these are coupled with the appropriate antibody-mediated effector mechanisms.

## 2. Results

### 2.1. SM Patients and Samples

A total of 206 BM samples from 116 SM patients and 40 controls (normal/reactive BM) were investigated. In each sample, CD117^hi^ CD45^int^ BMMC were analysed by flow cytometry for the expression of the distinct markers evaluated here: CD22, CD25, CD30, CD33, CD123, and FcϵRI.

### 2.2. Immuno-Phenotypic Characteristics of Normal/Reactive BMMC

MC from normal vs. reactive BM (control) samples showed overall similar immuno-phenotypic profiles (data not shown). Overall, reactivity for CD22 was found in the majority of control samples investigated (78%) (Figure 1A) with a median percentage of CD22^+^ MC of 89% (range: 0% to 100%) (Table 2). In turn, normal/reactive BMMC tested systematically positive for CD33 (100%) and FcεRI (100%) (Table 2), whereas CD25, CD30, and CD123 were found to be constantly absent on MC from normal/reactive BM (Table 2).

### 2.3. Immuno-Phenotypic Features of BMMC from SM Patients

CD22 was positive on BMMC from SM cases at similar percentages to those observed on normal/reactive BMMC (77% vs. 78%, *p* > 0.05) (Table 2), with similar median percentages of CD22^+^ cells (93% vs. 89% in normal BMMC, range: 0–100%, *p* > 0.05) (Table 2), CD22^+^ SM and normal/reactive BMMC also showed similar overall levels of expression for this marker (Figure 1A). In contrast, aberrant CD25 expression was found on BMMC from the great majority of SM patients (89% vs. 0% in normal BMMC, *p* = 0.001) (Table 2 and Figure 1B) with a relatively high median (92% vs. 0%, range: 0 to 100%, *p* = 0.001) percentage of CD25^+^ BMMC (Table 2). In addition, BMMC from SM patients also showed a greater rate of positivity for CD30 (63% vs. 5%, *p* = 0.001) and CD123 (38% vs. 3%, *p* = 0.001) than normal/reactive BMMC (Table 2, Figure 1C,E). This translated into aberrant CD25, CD30, and CD123 expression levels on BMMC from SM patients (vs. normal/reactive BMMC), both in terms of SI (*p* = 0.001) (Figure 1B,C,E) and mean fluorescence intensity (MFI) values (*p* = 0.001) (Table 2).

Similarly to normal/reactive MC, tumor BMMC from SM patients were also systematically positive for CD33 (100%, *p* > 0.05) on their surface membrane, at similarly high and homogenous levels as evaluated by the stain index (SI) (Figure 1D) and its coefficient of variation within individual cases (data not shown). In addition, reactivity for FcεRI was found in the great majority of SM cases (92% vs. 100% in normal BMMC; *p* = 0.05) (Table 2 and Figure 1F) with most BMMC (but not always all) typically showing FcεRI^+^ expression on their surface (95% vs. 100% in normal BMMC, *p* > 0.05) (Table 2).

Of note, no clearly strong (*σ* < 0.45) correlation was found between the expresion levels observed for the distinct cell surface proteins analyzed.

### 2.4. Immuno-Phenotypic Features of BMMC from Patients with Different Diagnostic and Prognostic Categories of SM

Similar percentages of CD22^+^ cases were found among all distinct diagnostic subtypes of SM (range: 75% to 90% of CD22^+^ cases) but MCL, which showed a lower rate of positivity (25%, *p* ≤ 0.03 vs. ISM groups) (Table 2). In line with this observation, the median percentage of CD22^+^ BMMC per patient, as well as CD22 expression levels per BMMC, were both similar among all patient groups except MCL, which showed significantly lower values (*p* ≤ 0.03 vs. all other groups of SM) (Table 2).

Regarding CD25, every BMMC from all ISM (100%) and ASM (100%) patients aberrantly expressed this marker on their surface, while it was present in only one third of WDSM and MCL cases, where it was also expressed at significantly lower percentages including a median of CD25^+^ BMMC of 44% (*p* ≤ 0.002 vs. ISM and *p* ≤ 0.005 vs. ASM subtypes) and 33% (*p* = 0.001 vs. ISM and *p* ≤ 0.004 vs. ASM), respectively (Table 2). Among all SM groups, ISM and ASM patients displayed the highest levels of CD25 per BMMC (*p* = 0.001 and *p* = 0.001 vs. BMMC from WDSM cases and *p* = 0.001 and *p* ≤ 0.005 vs. MCL patients) (Table 2 and Figure 1B).

Of note, CD30 expression was aberrantly high in a fraction of patients from every subtype of SM (*p* < 0.03) (Table 2). Despite this, in many SM patients, CD30 was not expressed by all tumor BMMC, the percentage of CD30^+^ BMMC in the different diagnostic subtypes of SM ranging from 0% in MCL patients to 94% in WDSM cases (*p* = 0.001), respectively (Table 2). Among the distinct diagnostic subgroups of SM, similar (but highly variable) CD30 expression levels per BMMC, were observed (*p* > 0.05) (Figure 1C).

Although CD33 was systematically expressed in all BMMC from SM patients (Table 2), among SM cases, BMMC from WDSM and ISM_MC_ patients displayed greater levels of CD33 (expressed per cell) than ISM_ML_ patients (*p* = 0.01 and *p* = 0.05), ASM (*p* = 0.001 and *p* = 0.002), and MCL (*p* = 0.005 and *p* = 0.007) (Figure 1D). In contrast, ASM patients showed lower levels of CD33 compared to ISM-AHN (*p* = 0.04), WDSM (*p* = 0.001), and ISM_MC_ (*p* = 0.002) cases (Figure 1D).

Additionally, CD123 only tested systematically positive in all BMMC from ASM-AHN cases (100%), the median percentage of CD123^+^ BMMC in the other subtypes of SM ranging from 17% in WDSM to 73% in ISM_ML_ (Table 2). In line with this, CD123 expression levels per BMMC were also higher among ASM-AHN cases vs. all other subtypes of SM (*p* ≤ 0.05). WDSM patients show the lowest expression levels (*p* ≤ 0.007 vs. all other groups) (Figure 1E).

Lastly, FcεRI was expressed in virtually all WDSM and ISM subtypes of SM (range: 89% to 100% cases) while showing a lower reactivity (*p* ≤ 0.03) in more advanced forms of SM (e.g., ASM and MCL), similarly to CD33. ISM_ML_ and ASM-AHN displayed intermediate FcεRI expression values (Figure 1F). Similarly, the percentage of FcεRI^+^ BMMC was high among all subtypes of SM except ASM and MCL (70% and 67%, respectively), that showed significantly lower percentages of FcεRI^+^ tumor BMMC (Table 2) together with significantly lower levels of this marker per MC (*p* ≤ 0.04 and *p* ≤ 0.03 vs. all groups of indolent SM) (Table 2 and Figure 1F).

## 3. Discussion

Currently, a high number of cytotoxic agents and targeted therapies exists for the treatment of advanced SM. Furthermore, several of them were proven to be of clinical benefit. Despite this, SM still remains an incurable disease with limited CR rates [63,64]. Thus, recent reports show increased overall response rates after treatment with midostaurin (60%) [10] particularly in advanced SM, but in the absence of any CR. Similarly, in a recent trial based on a small number (*n* = 10) of imatinib-treated SM patients, a high CR rate (40%) was reported, but only for patients carrying mutations in the extracellular region of *KIT* (K509I) (*n* = 3) or with wild-type *KIT* (*n* = 1), typically with a unique CD25^−^ and CD2^−^ phenotype [27], and that represent a minor fraction (≤5%) of all (indolent and advanced) SM cases [11]. Altogether, these results point out the need to search for new drugs and treatment modalities that might contribute to increased CR rates in SM and to improve the outcome of SM patients with advanced forms of the disease.

In recent years, immunotherapy has emerged as one of the most attractive and efficient cancer treatment strategies, particularly through antibody-mediated tumor-cell targeted therapies [20,65,66]. Despite the highly diverse antibody-based strategies that have been developed so far, with regard to the specific effector mechanisms used, e.g., from complement-mediated, macrophage-mediated, and/or NK cell-mediated antibody-dependent cytotoxicity approaches to chimeric antigen receptor (CAR)-cells, the efficacy of antibody-based immunotherapy requires the expression of enough amounts of the targeted protein on every individual (relevant) tumor cell [65,67,68,69,70,71,72].

Despite a great amount of data that has been generated about the phenotypic profiles of BMMC in SM, no study has specifically focused on the analysis of the patterns of expression of cell surface membrane proteins in every individual tumor cell that might be targeted by currently available antibody-based drugs approved for in vivo use in humans, in a per patient basis. In addition, no studies detailed analysis of the immunophenotypic profile of individual tumor MC within a large series of individual SM patients with distinct diagnostic subtypes of SM have been performed, based on standardized flow cytometry approaches, as seen here. Thus, we performed such an analysis for the first time, by investigating the expression of a broad panel of markers (CD22, CD25, CD30, CD33, CD123, and FcϵRI) that have been reported to be expressed on clonal BMMC from SM patients. Our major goal was to identify the best candidate marker(s) to be targeted by currently available antibody-based therapies approved by the FDA and/or EMA for their use in humans based on its pattern of expression on tumor MC. At this point, evaluation among the available antibody drugs among their specific effector mechanisms was not an aim of this study and requires further investigation.

Overall, our results confirm and extend on previous observations [24,25,73] regarding the existence of variable patterns of expression on BMMC from SM patients for all individual markers evaluated (i.e., CD22, CD25, CD30, CD33, CD123, and FcϵRI). Despite such heterogeneity, CD33 emerged as the only marker that is expressed on the cell surface of virtually every BMMC from individual patients of every subtype of SM even though greater expression levels were found in indolent SM and WDSM vs. advanced SM (i.e., ASM, SM-AHN and MCL). In turn, high levels of FcεRI and aberrant CD25 expression were found on tumor BMMC from the great majority of SM patients. However, FcεRI and CD25-negative BMMC were found in a fraction of the patients, particularly among advanced SM, CD25 is also absent in most WDSM patients.

In contrast, highly variable patterns of expression of CD22, CD30, and CD123 were observed both among distinct patients, and among different BMMC within a patient, pointing out these possibly as less suitable targets for antibody therapy. Despite this, CD123 should still probably be considered as a potentially useful target particularly in ASM-AHN cases where it tested systematically positive in the ASM component in parallel to a fraction of the AHN blast and tumor precursor cells (data not shown). Overall, these results confirm and extend on previous observations about the immunophenotypic profiles of BMMC in distinct diagnostic categories of SM. At the same time, they point out that CD33, and to a less extent also CD25 and FcϵRI, are very well-suited BMMC surface markers for targeted antibody therapies, while CD22, CD30, and CD123 are less suited cell surface targets except for CD123 in ASM-AHN cases. Further studies including quantitative flow cytometric methods with a panel of distinct reagents against CD33 are needed to confirm our results, since CD33 expression levels have also been associated previously with a further response to the anti-CD33 Mylotarg antibody in acute myeloid leukemia (AML) [74,75].

So far, only a few single case reports and small patient series that have been treated with antibody-targeted therapies have been reported in the literature. This includes four ASM patients [28] and one smoldering SM (SSM) case [50] treated with daclizumab (anti-CD25), four patients (three ASM and one ASM) treated with brentuximab-vedotin (anti-CD30) [29], one patient diagnosed with SM-AHN, one with MCL receiving gemtuzumab ozogamicin (anti-CD33) [26,27], and several SM patients with distinct subtypes of SM receiving omalizumab (anti-FcεRI) with significant improvement of symptoms but no objective reduction of tumor cell burden in most tested cases (CR: 52%, 11/21, major response: 33%, 7/21, partial response: 14%, 3/21) [54,55,56,57,58,59,60,61,62]. Moreover, some cutaneous mastocytosis (CM) patients have also been successfully treated with omalizumab with CR in 1/3 treated patients [76,77,78]. In contrast, no SM patients treated with anti-CD22 or anti-CD123 antibody-based immunotherapy have been reported so far.

In contrast to anti-CD30 and FcϵRI therapy, which were both associated with relatively poor responses in SM (e.g., overall response rate of 50% and 100% based on the improvement of symptoms but with only 52% partial responses, respectively) in the absence of any CR [29,54,55,56,57,58,59,60,61,62], anti-CD25, and anti-CD33 have shown more promising results [26,27,28,50]. Thus, a single dose of daclizumab, which is an anti-CD25 monoclonal antibody, was first administered as immunosuppressive therapy to an SSM patient who underwent a heart transplant. After treatment, a dramatic and sustained decrease in serum tryptase levels were observed, which suggests a significant response of her SSM [50]. Subsequently, a few ASM patients (*n* = 4) were also treated with daclizumab, with partial responses (25%). However, a limited effect on reducing the BMMC burden, which is associated with severe adverse effects (hypotension, tachycardia, and intense pain in back, hips, and femurs due to the massive release of intracellular MC granules) [28], suggests that daclizumab therapy might have some activity in a small subset of patients with advanced SM. In turn, sustained CR has been reported after anti-CD33 (gentuzumab ozogamicin) in an adult MCL patient who did not respond previously to imatinib, interferon, hydroxyurea, midostaurin, and cladribine [27], and in an SM associated with the AML child who finally died from relapsing AML [26]. In contrast, therapy with the humanized anti-CD30 brentuximab-vedotin antibody, that acts via a potent antimitotic agent (auristatin E) emerged as a potential candidate therapy for SM due to its ability (i) to induce apoptosis on neoplastic MC, (ii) to down-regulate of IgE-mediated histamine release, and (iii) to synergize with midostaurin to inhibit neoplastic MC growth in vitro [79]. However, once administered in vivo to SM patients, it was associated with severe adverse effects in the absence of CR [29]. Similarly, omalizumab—a humanized IgG kappa monoclonal antibody that acts via antibody-dependent cellular cytotoxicity (ADCC)—was initially shown to successfully control severe MC mediator-related symptoms in a CM patient [76,77,78]. This led to further administration of the drug to SM patients who displayed refractory symptoms of allergy in whom the drug proved to reduce the frequency of anaphylactic episodes in the absence of CR (or any significant effect on reducing the tumor MC burden), as reflected by the persistence of (stable) increased serum tryptase levels during treatment [54,55,56,57,58,59,60,61,62].

In summary, our results suggest that CD33 might potentially be the best candidate marker for BMMC surface protein targeted antibody therapy, while CD25 and FcεRI are potential alternative targets, whenever adequate effector cytotoxic mechanisms are used. In contrast, CD22, CD30, and CD123 emerge as less suited targets for immunotherapy in SM patients due to a lack of expression in a significant fraction of patients with ISM and particularly advanced forms of SM. The only exception is the potential utility of anti-CD123 therapy in ASM-AHN cases. However, our results are still preliminary since further in vitro and in vivo studies, as well as clinical trials, are required to confirm the potential efficacy and benefits of the specific antibody therapies.

## 4. Materials and Methods

### 4.1. Patients, Controls, and Samples

Overall, 206 BM samples from an identical number of adults—40 controls (6 normal BM samples from healthy subjects and 34 reactive BM samples, which correspond to patients undergoing BM aspiration due to a suspicion of a mast cell activation syndrome confirmed to be secondary to distinct underlying non-neoplastic conditions) and 166 patients diagnosed with SM at the reference centers of the Spanish Network on Mastocytosis (REMA, Mast Cell Unit, Hospital Virgen del Valle, Toledo, and, Cancer Research Center, Salamanca, Spain) were studied. According to the WHO criteria [3] and the pattern of BM involvement by the *KIT* mutation, patients were classified as: ISM, 111 cases (70 showed MC-restricted *KIT* mutation (ISM_MC_) and 41 multilineage involvement of BM haematopoiesis by the *KIT* mutation (ISM_ML_)), ASM, 10 patients, SM-AHN, 21 (12 ISM-AHN and 9 ASM-AHN), and MCL, 6 patients (aleukemic acute (*n* = 4) and chronic (*n* = 1) MCL variants, and 1 chronic leukemic MCL variant). The remaining 18 cases corresponded to WDSM variants of the disease [5]. All patients and controls gave their written informed consent to participate prior to entering the study, according to the Declaration of Helsinki, and the study was approved by the local Ethics Committee of the two participating centers.

### 4.2. Multi-Parameter Flow Cytometry Immuno-Phenotypic Studies

Multi-parameter flow cytometry immunophenotypic studies were performed on aspirated BM samples using a standardized stain-and-then-lyse direct immunofluorescence technique, as described elsewhere [24,80]. Additionally, 200 to 300 μL of BM per tube were stained for 30 min (room temperature) in the darkness with the antibody combinations described below. Then 2 mL/tube of the FACSLysing solution—Becton/Dickinson Biosciences (BD) San José, CA—diluted 1/10 (*v/v*) in distilled water was added and the stained samples incubated for another 10 min (room temperature) in the darkness. Afterward, non-lysed nucleated cells were centrifuged, washed twice in phosphate buffered saline (PBS), and re-suspended in 0.5 mL of PBS/tube. For staining purposes, two identical seven-color—pacific blue (PacB), pacific orange (PacO), fluorescein isothiocyanate (FITC), phycoerythrin (PE), peridinin-chlorophyll protein-cyanine 5.5 (PerCPCy5.5), phycoerythrin-cyanine 7 (PeCy7), allophycocyanine-Hilite^®^7 (APC-H7), which are combinations of monoclonal antibodies were used (Table 3), except for the APC-conjugated reagent that consisted of either CD33-APC or CD22-APC. For each individual marker (stained in ≥66 patient samples), the study only included cases in which the same antibody reagents had been used (antibody clones, fluorochrome conjugates, and manufacturer lots). Flow cytometry data acquisition was performed in a FACSCANTO II^TM^ instrument (BD), using FACS Diva software (BD). For data analysis, the INFINICYT^TM^ software (Cytognos SL, Salamanca, Spain) was used. Expression of individual markers on BMMC was assessed on gated CD117^hi^ CD45^int^ CD34^−^ events BMMC and expressed both as MFI values (arbitrary units scaled from 0 to 262,144) (Table 2) and SI (Figure 1). SI is the difference in signal width between the positive and the negative population divided by the signal spread of the negative population [81,82]. For each individual marker, the SI threshold for positivity was set at ≥1.5, after specifically gating on BMMC (Figure 2). For BMMC gating, a region was drawn to include all CD117^hi^ events after excluding cell debris and doublets in forward light scatter (FSC) vs. sideward light scatter (SSC) and FSC_area_ vs. FSC_width_ dot plots, respectively. Subsequently the CD117^hi^ events were gated on a CD45 vs. CD117 dot plot as CD45^int^/CD117^hi^ cells. The relative distribution of MC in BM samples analyzed by flow cytometry ranged from 0.001% to 0.5% in control vs. 0.0001% to 54% in SM cases.

### 4.3. Statistical Methods

For all phenotypic variables, median, range, mean, and standard deviation values, as well as 95% confidence intervals, were calculated. In order to determine the statistical significance of differences observed among groups, the Fisher χ^2^ exact test (for categorical variables) and the Mann Whitney-U and Kruskal-Wallis tests (for continuous variables) for comparisons between two or more than two groups were applied, respectively (SPSS 23 software, IBM, Armonk, NY, USA). The degree of correlation between different variables was assessed by the Spearman’s test (SPSS 23 software, IBM, Armonk, NY, USA). *p* values < 0.05 were considered to be associated with statistical significance.

## Figures and Tables

**Figure 1 ijms-20-00552-f001:**
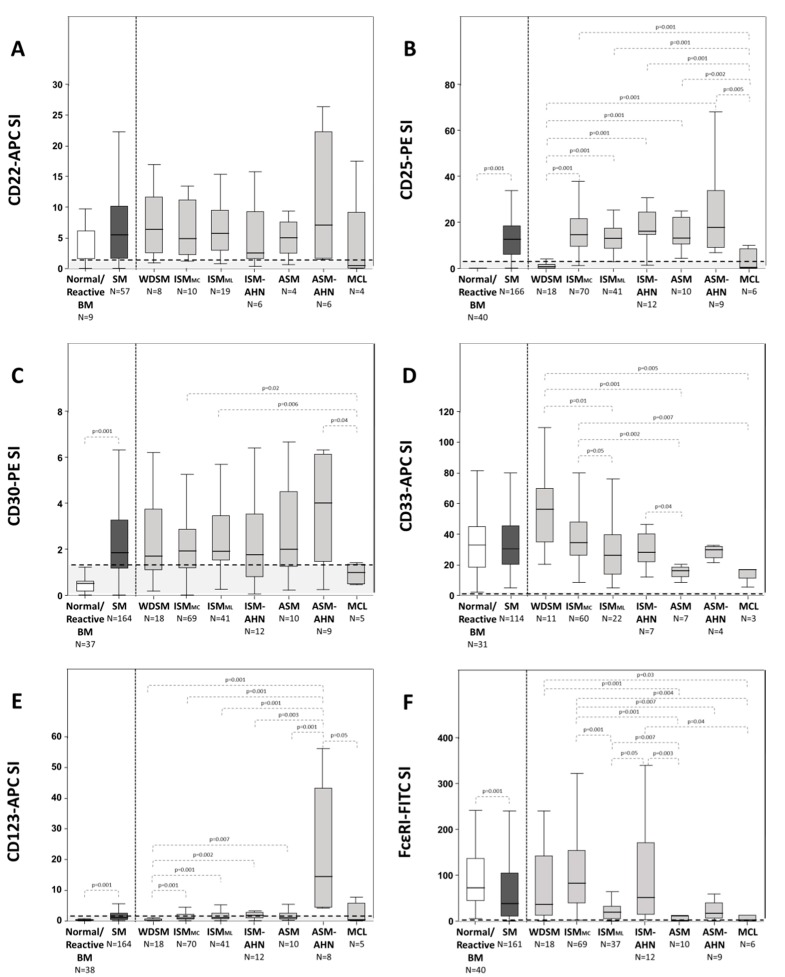
Immunophenotypic profile of BMMC from healthy subjects vs. SM patients classified according to the distinct diagnostic and prognostic categories of the disease. Results are expressed as stain index (SI, difference in signal width between the positive and the negative population divided by the signal spread of the negative population) for the CD22-APC (panel **A**), CD25-PE (panel **B**), CD30-PE (panel **C**), CD33-APC (panel **D**), CD123-APC (panel **E**) and FCεRI-FITC (panel **F**) markers. The SI threshold for positivity was set at ≥1.5 and is represented by horizontal dotted lines. *p*-values < 0.05 were considered to be associated with statistical significance (thin dotted lines). Mann Whitney-U test was used for comparisons between continuous variables (SI values) from normal/reactive individuals vs. SM patients or among the different subtypes of SM (SPSS 23 software, IBM, Armonk, NY, USA).

**Figure 2 ijms-20-00552-f002:**
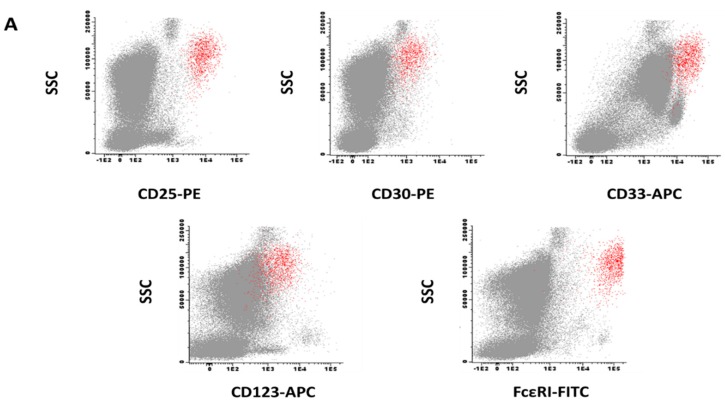
Representative bivariate dot plots illustrating the immuno-phenotype of pathological mast cells gated as CD117^hi^ cells (red events) and other residual bone marrow hematopoietic cells (grey events) in a patient with indolent systemic mastocytosis (panel **A**) and another case of mast cell leukemia (panel **B**). Abbreviations: SSC, sideward light scatter, PE, phycoerythrin, APC, allophycocyanine, FITC, fluorescein isothiocyanate.

**Table 1 ijms-20-00552-t001:** List of monoclonal antibodies directed against mast cell-associated cell surface markers that have been approved by the US Food and Drug Administration (FDA) and by the European Medicines Agency (EMA) for therapeutic use in humans or that are being evaluated in ongoing clinical trials.

Targeted Protein (Gene Symbol)	Monoclonal Antibody Name	Mechanism of Action	Year of Approval	Approved Indication(s)	References	ORR (CR) Rates in SM
**CD22** **(Siglec-2)**	Inotuzumab ozogamicin	Immunotoxin	2017 ^¥§^	B-cell precursor (BCP) acute lymphoblastic leukemia	[30,31]	-
	Moxetumomab pasudotox		2018 ^§^	Hairy cell leukemia	[32,33]	-
**CD25** **(IL2RA)**	Daclizumab	ADCC	1997 ^§#^ 1999 ^¥#^	Prophylaxis of acute organ rejection in renal transplantation	[34]	2/5 (40%) (0/5 (0%)) [28]
	Basiliximab		1998 ^¥§^	Prophylaxis of acute organ rejection in renal transplantation	[35,36]	-
	Daclizumab		2016 ^¥§#^	Relapsing forms of multiple sclerosis	[37,38,39,40]	-
**CD30** **(** **TNFRFS8)**	Brentuximab vedotin	Immunotoxin/ADCP	2011 ^§^ 2012 ^¥^	Hodgkin lymphoma/cutaneous lymphoma/peripheral T-cell lymphoma	[41,42,43,44,45,46]	2/4 50% (0/4 (0%)) [29]
**CD33** **(Siglec-3)**	Gemtuzumab ozogamicin	Immunotoxin	2000 ^#^ 2017 ^¥§^	Acute myeloid leukemia	[47,48,49]	(2/2 (100%)) [27,50]
**CD123** **(IL3RA)**	Talacotuzumab	ADCC	2018 ^*^	Acute myeloid leukemia	-	-
**FcεRI** **(High-affinity immunoglobulin ε heavy chain receptor I)**	Omalizumab	ADCC	2003 ^§^ 2005 ^¥^	Moderate to severe persistent asthma	[51,52,53]	10/21 (48%) (11/21 (52%)) [54,55,56,57,58,59,60,61,62]

# Marketing discontinued for the first approved indication. ^¥^ Year of approval in the European Union. ^§^ Year of approval in the United States of America. * Undergoing phase III clinical trials. Abbreviations: ADC, antibody-drug conjugate, ADCC, antibody-dependent cell-mediated cytotoxicity, ADCP, antibody-dependent cellular phagocytosis, CAR, chimeric antigen receptor, CR, complete remission, ORR, overall response rate.

**Table 2 ijms-20-00552-t002:** Expression profile for each immunophenotypic marker investigated on mast cells from normal/reactive BM vs. systemic mastocytosis patients distributed according to the diagnostic/prognostic category of the disease.

		Normal BM (*N* = 40)	SM (*N* = 166)	WDSM (*N* = 18)	ISM_MC_ (*N* = 70)	ISM_ML_ (*N* = 41)	ISM-AHN (*N* = 12)	ASM (*N* = 10)	ASM-AHN (*N* = 9)	MCL (*N* = 6)
**CD22**	*N*	9	57	8	10	19	6	4	6	4
	% positive patients	78	77	75	90	79	83	75	83	25
	*p*-value	-	-	-	¥ *p* = 0.02	¥ *p* = 0.03	-	-	-	§$ *p* ≤ 0.03
	% positive cells	89 (0–100)	93 (0–100)	100 (100–100)	100 (100–100)	100 (100–100)	83 (0–100)	75 (0–100)	100 (100–100)	50 (0–100)
	*p*-value	-	-	¥ *p* = 0.03	¥ *p* = 0.02	¶¥ *p* = 0.03	-	$ *p* = 0.03	-	†§$ *p* ≤ 0.03
	MFI	1333 (8–3431)	2629 (1–11,398)	2187 (192–6614)	2363 (424–5081)	2959 (348–11,398)	3086 (293–8612)	3521 (246–6760)	2841 (455–4435)	721 (1–2398)
	*p*-value	-	-	-	-	¥ *p* = 0.006	-	-	¥ *p* = 0.04	$‡ *p* ≤ 0.04
**CD25**	*N*	40	166	18	70	41	12	10	9	6
	% positive patients	0	89	33	99	100	92	100	100	33
	*p*-value	-	* *p* = 0.001	§$‖¶‡ *p* ≤ 0.002	†¥ *p* = 0.001	†¥ *p* = 0.001	†¥ *p* ≤ 0.009	†¥ *p* ≤ 0.003	†¥ *p* ≤ 0.004	§$‖¶‡ *p* ≤ 0.009
	% positive cells	0 (0–0)	92 (0–100)	44 (0–100)	100 (100–100)	100 (100–100)	100 (100–100)	100 (100–100)	100 (100–100)	33 (0–100)
	*p*-value	-	* *p* = 0.001	§$‖¶‡ *p* ≤ 0.005	†¥ *p* = 0.001	†¥ *p* = 0.001	†¥ *p* ≤ 0.002	†¥ *p* = 0.003	†¥ *p* ≤ 0.005	§$‖¶‡ *p* ≤ 0.004
	MFI	0 (0–0)	5331 (4–21,943)	471 (4–4159)	6448 (806–17,552)	5427 (2016–14,639)	6544 (2329–16,474)	6862 (1406–21,943)	5371 (2376–11,108)	1190 (21–4922)
	*p*-value	-	* *p* = 0.001	§$‖¶‡ *p* = 0.001	†¥ *p* = 0.001	†¥ *p* = 0.001	†¥ *p* = 0.001	†¥ *p* ≤ 0.005	†¥ *p* ≤ 0.008	§$‖¶‡ *p* ≤ 0.008
**CD30**	*N*	37	164	18	69	41	12	10	9	5
	% positive patients	5	63	61	61	76	58	70	67	0
	*p*-value	-	* *p* = 0.001	¥ *p* = 0.02	¥ *p* = 0.008	¥ *p* = 0.001	¥ *p* = 0.03	¥ *p* = 0.01	¥ *p* = 0.02	†§$‖¶‡ *p* ≤ 0.03
	% positive cells	0 (0–0)	88 (0–100)	94 (0–100)	87 (0–100)	90 (0–100)	83 (0–100)	90 (0–100)	89 (0–100)	0 (0–0)
	*p*-value	-	* *p* = 0.001	¥ *p* = 0.001	¥ *p* = 0.001	¥ *p* = 0.001	¥ *p* = 0.001	¥ *p* = 0.001	¥ *p* = 0.001	†§$‖¶‡ *p* = 0.001
	MFI	129 (1–427)	887 (0–6261)	632 (202–1677)	860 (0–3040)	993 (199–3038)	796 (103–2720)	1095 (74–2847)	1345 (202–6261)	297 (153–495)
	*p*-value	-	* *p* = 0.001	§$¥ *p* ≤ 0.02	†¥ *p* ≤ 0.02	†¥ *p* ≤ 0.01	-	-	¥ *p* = 0.02	†§$‡ *p* ≤ 0.02
**CD33**	*N*	31	114	11	60	22	7	7	4	3
	% positive patients	100	100	100	100	100	100	100	100	100
	*p*-value	-	-	-	-	-	-	-	-	-
	% positive cells	100 (100–100)	100 (100–100)	100 (100–100)	100 (100–100)	100 (100–100)	100 (100–100)	100 (100–100)	100 (100–100)	100 (100–100)
	*p*-value	-	-	-	-	-	-	-	-	-
	MFI	8163 (2225–16,377)	19,667 (2308–53,009)	16,642 (5774–50,767)	22,673 (4926–53,009)	20,576 (2922–44,241)	15,566 (5441–24,537)	10,063 (2698–18,730)	8447 (4022–14,916)	10,917 (2308–23,681)
	*p*-value	-	* *p* = 0.001	§ *p* = 0.03	†¶‡ *p* ≤ 0.03	¶ *p* = 0.02	-	§$ *p* ≤ 0.02	§ *p* = 0.002	-
**CD123**	*N*	38	164	18	70	41	12	10	8	5
	% positive patients	3	38	11	37	32	67	40	100	40
	*p*-value	-	* *p* = 0.001	§‖‡ *p* ≤ 0.03	†‡ *p* ≤ 0.03	‖‡ *p* ≤ 0.03	†$ *p* ≤ 0.03	‡ *p* = 0.007	†§$¶¥ *p* ≤ 0.01	‡ *p* = 0.01
	% positive cells	0 (0–0)	64 (0–100)	17 (0–100)	66 (0–100)	73 (0–100)	70 (0–100)	70 (0–100)	100 (100–100)	40 (0–100)
	*p*-value	-	* *p* = 0.001	§$‖¶‡ *p* = 0.001	†‡ *p* ≤ 0.05	† *p* = 0.001	† *p* = 0.001	† *p* = 0.001	†§¥ *p* ≤ 0.05	‡ *p* = 0.01
	MFI	84 (8–221)	1340 (3–33 861)	214 (3–2023)	1024 (17–5425)	962 (15–4045)	3685 (24–33,861)	939 (94–3220)	5825 (755–18,421)	903 (14–3085)
	*p*-value	-	* *p* = 0.001	§$‖¶‡ *p* = 0.001	†‡ *p* = 0.001	†‡ *p* = 0.001	†‡ *p* ≤ 0.01	†‡ *p* ≤ 0.003	†§$‖¶¥ *p* ≤ 0.03	‡ *p* = 0.03
**FcεRI**	*N*	40	161	18	69	37	12	10	9	6
	% positive patients	100	92	94	99	89	100	60	89	67
	*p*-value	-	-	¶ *p* = 0.02	$¶¥ *p* ≤ 0.03	§¶ *p* = 0.03	¶¥ *p* ≤ 0.03	†§$‖ *p* ≤ 0.03	-	§‖ *p* ≤ 0.03
	% positive cells	100 (100–100)	95 (0–100)	94 (0–100)	100 (100–100)	95 (0–100)	100 (100–100)	70 (0–100)	100 (100–100)	67 (0–100)
	*p*-value	-	-	§ *p* = 0.05	†¶¥ *p* ≤ 0.05	¶¥ *p* = 0.001	¶¥ *p* = 0.04	§$‖ *p* ≤ 0.04	-	§$‖ *p* ≤ 0.03
	MFI	29,073 (2181–81,199)	49,387 (14–239,335)	40,350 (110–183,969)	78,922 (1271–239,335)	24,084 (73–162,320)	62,685 (1938–177,643)	4858 (176–26,255)	11,084 (680–36,688)	9909 (14–52,514)
	*p*-value	-	-	§¶¥ *p* ≤ 0.02	†$¶‡¥ *p* ≤ 0.002	§‖¶ *p* ≤ 0.04	$¶‡¥ *p* ≤ 0.04	†§$‖ *p* ≤ 0.005	§‖ *p* ≤ 0.04	†§‖ *p* ≤ 0.02

Results expressed as median values and range between brackets. Abbreviations: BM, bone marrow, WDSM, well-differentiated SM, ISM_MC_, indolent SM with *KIT* mutation restricted to mast cells, ISM_ML_, ISM with multi-lineal *KIT* mutation, ISM-AHN, ISM associated with a clonal non-mast cell lineage hematopoietic disease, ASM, aggressive SM, ASM-AHN, ASM associated with a clonal non-mast cell lineage hematopoietic disease, MCL, mast cell leukemia, SI, stain index is the difference in signal width between the positive and the negative population divided by the signal spread of the negative population, and MFI, median fluorescence intensity (arbitrary values scaled from 0 to 222 144). *p*-values < 0.05 were considered to be associated with statistical significance for comparisons against: * normal/reactive BM, ^†^ WDSM, ^§^ ISM_MC_, ^$^ ISM_ML_, ^‖^ ISM-AHN, ^¶^ ASM, ^‡^ ASM-AHN, and ^¥^ MCL cases.

**Table 3 ijms-20-00552-t003:** List of fluorochrome-conjugated monoclonal antibodies used in this study to characterize BMMCs from SM patients and normal/reactive BM.

Marker	Clone	Fluorochrome	Source	Specificity
**CD22**	S-HCL-1	APC	BD Biosciences ^*^	Sialic acid-binding Ig-like lectin 2 (Siglec-2)
**CD25**	2A3	PE	BD Biosciences ^*^	Interleukin-2 receptor, subunit α
**CD30**	Ber-H8	PE	BD Biosciences ^*^	Member 8 of the tumor necrosis factor receptor superfamily
**CD33**	P67.6	APC	BD Biosciences ^*^	Sialic acid-binding Ig-like lectin 3 (Siglec-3)
**CD123**	AC145	APC	Miltenyi Biotec ^†^	Interleukin-3 receptor, subunit α
**IgE**	Polyclonal	FITC	Invitrogen ^§^	High-affinity immunoglobulin ε heavy chain receptor I

Abbreviations: APC, allophycocyanine, PE, phycoerythrin, FITC, fluorescein isothiocyanate. ^*^ Becton/Dickinson Biosciences (BD, San José, CA, USA), ^†^ Miltenyi Biotec (Bergisch Gladbach, Germany), ^§^ Invitrogen (Carlsbad, CA, USA).

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
