# Peer review of "Bone Marrow Mast Cell Antibody-Targetable Cell Surface Protein Expression Profiles in Systemic Mastocytosis"

_ijms, 2019, doi:10.3390/ijms20030552_

Round 1
Reviewer 1 Report
As the authors mentioned, they provided for the first time, detailed information about the patterns of expression on BMMC from quite significant amount of Systemic mastocytosis patients objectively. With limited published work with similar findings, those information are very useful for basic research and clinical treatment directions in Systemic mastocytosis. Very informative study.
Author Response
General comment.- As the authors mentioned, they provided for the first time, detailed information about the patterns of expression on BMMC from quite significant amount of Systemic mastocytosis patients objectively. With limited published work with similar findings, those information are very useful for basic research and clinical treatment directions in Systemic mastocytosis. Very informative study.
Answer to the general comment.- We thank the reviewer for the executive summary about the paper contents and his/her positive criticisms on the work presented, and we have nothing to add in this regard.
Reviewer 2 Report
ijms-409428
“Bone marrow mast cell antibody-targetable cell surface protein expression profiles in systemic mastocytosis” by Dasilva-Freire et al.
In this paper, the authors examined the expression of 6 mast cell markers on BMMC from systemic mastcytoma patients. They found only CD33 was expressed in all cases, indicating that CD33 was the best target for therapeutics.
The results were very important ovservation for clinical treatment. However, the paper to IJMS, the authors need more examination.
Major points:
1. In table, the authors listed antibodies for therapeutic use. How about their response rates? Is Anti-CD33 show the highest rate? The authors should compared the clinical data to the present results.
2. Is there any relevant between the expression level of CD33 and the efficacy of the antibody. The cells were the mixture of CD33-high expressed cells and relatively low expressed cells.
3. Is there any difference in the efficacy of imatinib among the different marker expressing patients. Is there any possibility that these markers are use for personalized treatment with imatinib?
Author Response
General comment.- In this paper, the authors examined the expression of 6 mast cell markers on BMMC from systemic mastcytoma patients. They found only CD33 was expressed in all cases, indicating that CD33 was the best target for therapeutics. The results were very important observation for clinical treatment. However, the paper to IJMS, the authors need more examination.
Answer to the general comment.- We thank the reviewer for the brief to-the-point summary of the paper findings. Following the comments raised by the reviewer, the paper has been modified as described below, in order to reach the standards for publication in IJMS.
Major comments:
Comment 1.- In table, the authors listed antibodies for therapeutic use. How about their response rates? Is Anti-CD33 show the highest rate? The authors should compared the clinical data to the present results.
Answer to comment 1.- Information about the response rates observed for the distinct antibodies used in mastocytosis has been added in a new column in Table 1. In addition, this data is discussed in detail in the Discussion section of the revised manuscript.
Comment 2.- Is there any relevant between the expression level of CD33 and the efficacy of the antibody. The cells were the mixture of CD33-high expressed cells and relatively low expressed cells.
Answer to comment 2.- A more detailed description of the pattern of expression of CD33 on BM MC from mastocytosis patients, including intra-individual heterogeneity, is now provided in the Results section of the revised manuscript, in line with the comment of the reviewer.
Comment 3.- Is there any difference in the efficacy of imatinib among the different marker expressing patients. Is there any possibility that these markers are use for personalized treatment with imatinib?
Answer to comment 3.- A new sentence has been added in the discussion section of the revised manuscript in which the potential and known correlates between cell surface marker expression and response to imatinib are discussed.
Reviewer 3 Report
Expression levels of six surface proteins on bone marrow mast cell samples from 166 patients with systemic mastocytosis were examined by flow cytometry. The antigens for this expression analysis were selected from the targets of monoclonal antibodies that are in clinical use against myeloid malignancies.
1) A comparative assessment of the surface expression on systemic mastocytosis samples of selected antigens may be of value to guide the potential repurposing of existing monoclonal antibodies that already have been approved for other indications. Without any attempt to evaluate the efficacy of the corresponding target antibodies, e.g. in ADCC assays of cytotoxicity assays of conjugated antibodies, the study is bound to be rated as preliminary with limited possibilities for publication in high impact journals.
2) The value of the study lies in the relatively high number of examined mastocytosis bone marrow samples that are not easily accessible. It is not clear, for which reasons the six target antigens assessed in this study had been selected, for instance why CD22 (SIGLEC2, an ITIM containing receptor on B cells and NOT mentioned in ref. 22) is included and e.g. CD44 is not. Table 1 is not very helpful in this context. It would be better to have a list of the antigens, enhanced by gene symbols and an indication of protein function (similarly as now in Table 3), plus a short mentioning of representative corresponding monoclonal antibodies targeting the respective surface proteins and the indications they have been approved for. The mentioned reagents must be restricted to clinically approved antibodies with relevance in this context. Candidate antibodies and further details of their development may be shown in the supplement, if at all in this manuscript. Moreover, reference 20 is about chimeric T cell receptors not about monoclonal antibodies.
3) Using different antibodies with different labels, calibration is required to quantitatively compare the surface protein density on target cells, i.e. numbers of the different antigens per cell. Without calibration MFI cannot yield the density of different antigens (when determined with reagents showing different affinity, labeling density and signal strength), but only compare the distribution among samples. This limitation must be mentioned.
4) Since the Methods section is at the end of the manuscript, it is necessary to include already in the Results section sufficient methodological detail for the reader to follow the procedures that led to the results.
5) The description of flow cytometric procedures needs to be thoroughly improved with a detailed outline of the gating strategy and of the two sets of antibodies for MFI determination. While fluorescently labelled antibodies raised against the surface proteins were used in most cases, IgE was used as ligand to detect Fc-epsilon receptor. Are “reactive” bone marrow samples derived from SM patients and how are they distinguished from SM samples?
It may be appropriate to use some sort of staining index instead of MFI differences, but the indicated calculation of staining index appears not to make sense. On p. 12 line 47 it is defined as the MFI difference “divided by 2 SD of the MFI values of the negative population”. Since negative population means unstained sample, its MFI is the background fluorescence, with usually a very low value and even lower SD, which thus could approach zero.
Author Response
General comment.- Expression levels of six surface proteins on bone marrow mast cell samples from 166 patients with systemic mastocytosis were examined by flow cytometry. The antigens for this expression analysis were selected from the targets of monoclonal antibodies that are in clinical use against myeloid malignancies.
Answer to the general comment.- The reviewer provides a precise summary of the work done and we have nothing to add in this regard.
Specific comments:
Comment 1.- A comparative assessment of the surface expression on systemic mastocytosis samples of selected antigens may be of value to guide the potential repurposing of existing monoclonal antibodies that already have been approved for other indications. Without any attempt to evaluate the efficacy of the corresponding target antibodies, e.g. in ADCC assays of cytotoxicity assays of conjugated antibodies, the study is bound to be rated as preliminary with limited possibilities for publication in high impact journals.
Answer to comment 1.- We have toned down the conclusions of the study and highlighted the need for further in vitro and patient studies, in order to definitively evaluate the efficacy of the studied target antibodies. In addition, new information has been added in table 1 about the overall response rates observed with the investigated antibody targets in systemic mastocytosis patients.
Comment 2.- The value of the study lies in the relatively high number of examined mastocytosis bone marrow samples that are not easily accessible. It is not clear, for which reasons the six target antigens assessed in this study had been selected, for instance why CD22 (SIGLEC2, an ITIM containing receptor on B cells and NOT mentioned in ref. 22) is included and e.g. CD44 is not. Table 1 is not very helpful in this context. It would be better to have a list of the antigens, enhanced by gene symbols and an indication of protein function (similarly as now in Table 3), plus a short mentioning of representative corresponding monoclonal antibodies targeting the respective surface proteins and the indications they have been approved for. The mentioned reagents must be restricted to clinically approved antibodies with relevance in this context. Candidate antibodies and further details of their development may be shown in the supplement, if at all in this manuscript. Moreover, reference 20 is about chimeric T cell receptors not about monoclonal antibodies.
Answer to comment 2.- Following the comment of the reviewer protein and gene symbols, as well as antibody clones have been added in Table 1 of the revised manuscript. In addition, a new sentence has been added in the Discussion section of the revised manuscript about the reason for selecting the markers evaluated and not others (e.g. CD44). Also, reference 20 has been changed.
Comment 3.- Using different antibodies with different labels, calibration is required to quantitatively compare the surface protein density on target cells, i.e. numbers of the different antigens per cell. Without calibration MFI cannot yield the density of different antigens (when determined with reagents showing different affinity, labeling density and signal strength), but only compare the distribution among samples. This limitation must be mentioned.
Answer to comment 3.- Following the indication of the reviewer, a new sentence has been added in the Discussion section of the revised manuscript in which, the limitations associated with the use of MFI values to assess antigen expression levels on individual cells are highlighted.
Comment 4.- Since the Methods section is at the end of the manuscript, it is necessary to include already in the Results section sufficient methodological detail for the reader to follow the procedures that led to the results.
Answer to comment 4.- We thank the reviewer for highlighting the need to include some critical information about the methods used at the beginning of the Results section. Thus, a new section has been added in the first part of the Results section where essential patient and methods information are provided now for easy reading of the contents of the manuscript in the format of the journal.
Comment 5.- The description of flow cytometric procedures needs to be thoroughly improved with a detailed outline of the gating strategy and of the two sets of antibodies for MFI determination. While fluorescently labelled antibodies raised against the surface proteins were used in most cases, IgE was used as ligand to detect Fc-epsilon receptor. Are “reactive” bone marrow samples derived from SM patients and how are they distinguished from SM samples? It may be appropriate to use some sort of staining index instead of MFI differences, but the indicated calculation of staining index appears not to make sense. On p. 12 line 47 it is defined as the MFI difference “divided by 2 SD of the MFI values of the negative population”. Since negative population means unstained sample, its MFI is the background fluorescence, with usually a very low value and even lower SD, which thus could approach zero.
Answer to comment 5.- In the last paragraph of the introduction section the criteria used to include markers in the study are now clearly described. Detailed information about the gating strategy used to identify mast cells is now provided in the methods section of the revised manuscript. A new sentence has also been added in the methods section of the revised manuscript about the anti-IgE receptor I antibody used. Finally, results about the stain index are also provided now in Supplementary materials, following the suggestion of the reviewer.
Reviewer 4 Report
This is a comprehensive study showing the expression profiles of targetable surface markers on systemic mastocytosis BMMCs. The results demonstrate that, while several surface markers are elevated in SM, CD33 is universally expressed on all mast cells. Generally, the article is well written and, although the markers of interest have previously been described for SM, the present study has used a larger cohort of patient samples (including SM subtypes) and the authors have deliberated their findings with respect to potential current monoclonal antibody therapeutic approaches. I have no major concerns except that table 2 should be replaced with table S1 (and the latter deleted), since they essentially show the same information (though table 1 seems to have contains errors, such as omitting CD33 labels).
Minor points:
Table 1 need reformatting. It is also unclear why 3 different shades are used.
Inconsistency wrt BM MC or BMMC
Author Response
General comment.- This is a comprehensive study showing the expression profiles of targetable surface markers on systemic mastocytosis BMMCs. The results demonstrate that, while several surface markers are elevated in SM, CD33 is universally expressed on all mast cells. Generally, the article is well written and, although the markers of interest have previously been described for SM, the present study has used a larger cohort of patient samples (including SM subtypes) and the authors have deliberated their findings with respect to potential current monoclonal antibody therapeutic approaches.
Answer to the general comment.- We thank the reviewer for the overall very positive comments about the work done and its utility, and we have nothing to add in this regard.
Major Comment .- I have no major concerns except that table 2 should be replaced with table S1 (and the latter deleted), since they essentially show the same information (though table 1 seems to have contains errors, such as omitting CD33 labels).
Answer to the major comment.- Table 2 has been deleted and replaced in the main body of the manuscript by a modified version of supplementary table 1, following the indication of the reviewer. The text of the revised manuscript has also been modified accordingly.
Minor points:
Point 1.- Table 1 need reformatting. It is also unclear why 3 different shades are used.
Answer to point 1.- Table 1 has been modified and the shades deleted, following the comments of the reviewer.
Point 2.- Inconsistency wrt BM MC or BMMC.
Answer to point 2.- The abbreviation for bone marrow mast cells is now consistently written as BMMC.
Round 2
Reviewer 2 Report
The authors revised this muascript well according to reviewers' comments.
Author Response
We thank the reviewer for acknowledging the effort done in the first revised version of the manuscript to address the comments he/she raised, and we have nothing to add in this regard.
Reviewer 3 Report
Expression levels of six surface proteins on bone marrow mast cell samples from 166 patients with systemic mastocytosis were examined by flow cytometry. The antigens for this expression analysis were selected from the targets of monoclonal antibodies that are in clinical use against myeloid malignancies, with the goal of guiding the potential repurposing of existing monoclonal antibodies that already have been approved for other indications.
The authors have provided point by point responses to reviewer queries, which solved some minor points, but did not sufficiently address the major criticized items. The manuscript still must be thoroughly changed to correctly render the described observations and justify the conclusions. The amended manuscript is difficult to evaluate owing to excessive mark-ups that refer to minor formatting issues. Please provide more user-friendly mark-ups during the next revision, e.g. by accepting format changes.
1) As a courtesy to readers, the investigated surface antigens must be listed consistently in the same order, e.g. alphanumerically, throughout the manuscript, i.e. in the abstract, Tables, Figures and text.
2) The current listing of monoclonal antibodies in the order of their development in Table 1 is confusing and not acceptable in this context. It must be replaced by a completely different Table that is ordered according to the investigated surface antigens. Each antigen may be accompanied by one or very few examples of clinically used antibodies, but must not contain the excessive number of developed reagents listed in the current version. Only reagents with formal approval should be selected for the compilation at this point. Trade names, the developing companies and approval or description dates are irrelevant in the present context. Moreover, in most cases, the indicated years do not indicate formal approval as stated in the column heading, but some other date in the development. In a suitable version of this Table, the description of the antibodies for the most part can be accomplished by references to the literature and can be largely limited to their mechanism of action (i.e. immunotoxin or cellular cytotoxicity) and the indications, the reagents are used for.
3) It is not clearly indicated, how the investigated samples were derived from bone marrow aspirates. On page 6, lines 1-4, it is mentioned that all samples were FACS sorted, but this statement is late in the description of the procedure and does not name the parameters used for sorting. The present description does not allow readers to follow, how exactly the samples have been prepared, even if they refer to the indicated citations. Typical ranges of the content of mast cells in bone marrow aspirates and FACS-sorted samples must be indicated for healthy donors and systemic mastocytosis patients. What are typical enrichment factors during FACS sorting? What was the sample preparation for the examples shown in Fig. 2?. How are “reactive” bone marrow samples from SM patients defined? A thorough description of sample work-up is required in the Methods section.
4) The main results are presented in box plots of stain index values in Fig. 1. Therefore it is obvious, that a valid definition of the stain index must be given along with an exact description how these values were derived. Without proper definition the use of the stain index is pointless and must be left out, which requires presentation of the results in a different measure, e.g. MFI.
5) The statistics in Fig. 1 is insufficiently explained. The use of symbols in the right part of the diagrams is confusing and should be replaced by lines connecting the boxes.
Author Response
General comment.- Expression levels of six surface proteins on bone marrow mast cell samples from 166 patients with systemic mastocytosis were examined by flow cytometry. The antigens for this expression analysis were selected from the targets of monoclonal antibodies that are in clinical use against myeloid malignancies, with the goal of guiding the potential repurposing of existing monoclonal antibodies that already have been approved for other indications.
The authors have provided point by point responses to reviewer queries, which solved some minor points, but did not sufficiently address the major criticized items. The manuscript still must be thoroughly changed to correctly render the described observations and justify the conclusions. The amended manuscript is difficult to evaluate owing to excessive mark-ups that refer to minor formatting issues. Please provide more user-friendly mark-ups during the next revision, e.g. by accepting format changes.
Answer to the general comment.- We thank the reviewer for acknowledging the effort done to modify the manuscript following the comments and criticisms raised during the first revision round. We apologise for not having adequately addressed some major concerns, which we hope have been appropriately addressed now in this second revised version of the manuscript, as detailed below. Track changes related to format have been omitted in the submitted marked version of the manuscript, following the indication of the reviewer.
Specific comments:
Comment 1.- As a courtesy to readers, the investigated surface antigens must be listed consistently in the same order, e.g. alphanumerically, throughout the manuscript, i.e. in the abstract, Tables, Figures and text.
Answer to comment 1.- The investigated surface antigens are now listed consistently in the same order throughout the manuscript, including the Tables, Figures and the main text of the manuscript.
Comment 2.- The current listing of monoclonal antibodies in the order of their development in Table 1 is confusing and not acceptable in this context. It must be replaced by a completely different Table that is ordered according to the investigated surface antigens. Each antigen may be accompanied by one or very few examples of clinically used antibodies, but must not contain the excessive number of developed reagents listed in the current version. Only reagents with formal approval should be selected for the compilation at this point. Trade names, the developing companies and approval or description dates are irrelevant in the present context. Moreover, in most cases, the indicated years do not indicate formal approval as stated in the column heading, but some other date in the development. In a suitable version of this Table, the description of the antibodies for the most part can be accomplished by references to the literature and can be largely limited to their mechanism of action (i.e. immunotoxin or cellular cytotoxicity) and the indications, the reagents are used for.
Answer to comment 2.- The order of the markers displayed in Table 1 has been changed following the indication of the reviewer. In addition, the contents of Table 1 have also been condensed, following the suggestion of the reviewer.
Comment 3.- It is not clearly indicated, how the investigated samples were derived from bone marrow aspirates. On page 6, lines 1-4, it is mentioned that all samples were FACS sorted, but this statement is late in the description of the procedure and does not name the parameters used for sorting. The present description does not allow readers to follow, how exactly the samples have been prepared, even if they refer to the indicated citations. Typical ranges of the content of mast cells in bone marrow aspirates and FACS-sorted samples must be indicated for healthy donors and systemic mastocytosis patients. What are typical enrichment factors during FACS sorting? What was the sample preparation for the examples shown in Fig. 2?. How are “reactive” bone marrow samples from SM patients defined? A thorough description of sample work-up is required in the Methods section.
Answer to comment 3.- The description of the immunophenotypic procedures techniques applied to the bone marrow aspirated samples, including FACS-sorting, has now been expanded in the material and methods section of the revised manuscript, following the comment of the reviewer.
Comment 4.- The main results are presented in box plots of stain index values in Fig. 1. Therefore it is obvious, that a valid definition of the stain index must be given along with an exact description how these values were derived. Without proper definition the use of the stain index is pointless and must be left out, which requires presentation of the results in a different measure, e.g. MFI.
Answer to comment 4.- Stain index values are considered as a more appropriate measure of the intensity of fluorescence expression levels than mean fluorescence intensity values, since they correct for differences in voltages/background fluorescence intensity of negative (control) cell populations. The standard formula used to calculate stain indices is now clearly defined in the material and methods section of the revised manuscript, as well as in the legend to Figure 1 and the footnote in Table 1. In addition, new references have been added to support the appropriate usage of stain index vs MFI values. In any case, MFI values are also specified in Table 1, in addition to SI.
Comment 5.- The statistics in Fig. 1 is insufficiently explained. The use of symbols in the right part of the diagrams is confusing and should be replaced by lines connecting the boxes.
Answer to comment 5.- A new sentence has been added in the legend to Figure 1, about the specific statistical tests applied for the comparisons displayed in this Figure. In addition, lines connecting the distinct box plots, are now used in Figure 1, instead of the symbols, following the indication of the reviewer.
Round 3
Reviewer 3 Report
Expression levels of six surface proteins on bone marrow mast cell samples from 166 patients with systemic mastocytosis were evaluated by flow cytometry with subsequent subgroup analysis to guide the potential repurposing of existing monoclonal antibodies that already have been approved or investigated for other indications.
In the present third version of this manuscript criticized issues have been addressed to the degree that a full appreciation of the described work becomes possible. However, further changes are still required. Contrary to what is ascertained in the author response, most of the raised issues were only partially resolved in the manuscript. Again a version with distracting formatting information and comments not intended for peer review was delivered. The further necessary changes are listed below in the same order as in the review of the second manuscript version.
1) Thank you for presenting the investigated surface antigens consistently in the same order. This makes it easier for readers to compare the results. The order of surface antigens still needs to be changed in the supplemental Table.
2) After two revisions, Table 1 starts to serve the purpose of linking the investigated surface antigens with available therapeutic agents, since the Table was rearranged and the excessive number of listed agents was reduced. While the exact IgG subtype is of little relevance in this context, the previously suggested consistent distinction of antibodies as agents acting via cellular cytotoxicity or immunotoxins (or bifunctional T cell engager, if included) is still missing. As mentioned before, references to admission studies are much more informative for readers than the years of admission in the US or EU. In addition, the remission rates in the last column must be linked to the corresponding references. Since CD123 is also among the investigated surface antigens, one or a couple of clinically developed antibodies directed against this antigen should be included in Table 1, even if they have not yet attained formal approval.
3) Thank you for the clarifications of the immunophenotypical procedures. Since KIT mutation analysis is not within the scope of the present work, would it not be better to eliminate the corresponding procedures, including FACS sorting, (i.e. chapter 4.3) from the Methods section, to avoid confusion? Although the samples shown in Fig. 3 have a widely different content of CD117hi cells, it still would be interesting to know the typical ranges of the content of mast cells in bone marrow aspirates from healthy donors and systemic mastocytosis patients.
4) The source of confusion in the definition of the stain index was the term “standard deviation”, which is ambiguous, since here it means signal spread, but perhaps more frequently it refers to inter-sample variation. Therefore I propose to use in the definition the up-dated terminology used by Maecker & Trotter, 2008, Nature Methods, saying something like “Stain index is the difference in signal width between the positive and the negative population divided by the signal spread of the negative population.” I think that this Nature Methods article, would be more appropriate than reference 61, since it efficiently explains the staining index by means of an illustration.
5) Thanks for replacing the symbols in Fig. 2 by connecting lines. In my opinion, this shows the statistical differences between the populations in the box plot presentation in a more user-friendly manner. I do not think that it is necessary to repeat the stain index values and their p-values in a Table, particularly not in the main text.
Additional minor remarks:
1) It is more exact and therefore preferable to speak of “six analyzed surface proteins” rather than a “relatively large panel” (lines 27/p. 1 and 38/p. 3).
2) The spelling of Gemtuzumab ozogamycin in Table 1 must be corrected.
3) “Despite” in lines 38/p.11 and 32/p.12 must be replace by “although”.
4) The reference list could be shortened.
Author Response
General comment.- Expression levels of six surface proteins on bone marrow mast cell samples from 166 patients with systemic mastocytosis were evaluated by flow cytometry with subsequent subgroup analysis to guide the potential repurposing of existing monoclonal antibodies that already have been approved or investigated for other indications. In the present third version of this manuscript criticized issues have been addressed to the degree that a full appreciation of the described work becomes possible. However, further changes are still required. Contrary to what is ascertained in the author response, most of the raised issues were only partially resolved in the manuscript. Again a version with distracting formatting information and comments not intended for peer review was delivered. The further necessary changes are listed below in the same order as in the review of the second manuscript version.
Answer to the general comment.- We thank the reviewer for the executive summary of the paper and apologise for not having addressed adequately/completely his previous comments and sugestions. We have now addressed the further comments raised on the second revised version (R2) of the manuscript and hope have committed with all issues raised in this second set of comments and concerns, as explained below in the point-by-point list of the answers to the comments of the reviewer. We hope the clean version highlighting in yellow the changes introduced is clearer for the reviewer and facilitates the reading of the manuscript.
Comments:
Comment 1.- Thank you for presenting the investigated surface antigens consistently in the same order. This makes it easier for readers to compare the results. The order of surface antigens still needs to be changed in the supplemental Table.
Answer to comment 1.- Supplemental Table 1 has been already deleted in the previous version of the manuscript.
Comment 2.- After two revisions, Table 1 starts to serve the purpose of linking the investigated surface antigens with available therapeutic agents, since the Table was rearranged and the excessive number of listed agents was reduced. While the exact IgG subtype is of little relevance in this context, the previously suggested consistent distinction of antibodies as agents acting via cellular cytotoxicity or immunotoxins (or bifunctional T cell engager, if included) is still missing. As mentioned before, references to admission studies are much more informative for readers than the years of admission in the US or EU. In addition, the remission rates in the last column must be linked to the corresponding references. Since CD123 is also among the investigated surface antigens, one or a couple of clinically developed antibodies directed against this antigen should be included in Table 1, even if they have not yet attained formal approval.
Answer to comment 2.- The column of the Ig isotype of the therapeutic reagents listed in table 1 has been deleted and replaced by a new column in which their mechanism of action (e.g. via cellular cytotoxicity, immunotoxins). In addition to the remission rates, the references to the most relevant papers (including those used for FDA or EMA approval for drugs that have reached approval) related with each therapeutic reagent listed, have been added in a new column in Table 1.
Comment 3.- Thank you for the clarifications of the immunophenotypical procedures. Since KIT mutation analysis is not within the scope of the present work, would it not be better to eliminate the corresponding procedures, including FACS sorting, (i.e. chapter 4.3) from the Methods section, to avoid confusion? Although the samples shown in Fig. 3 have a widely different content of CD117hi cells, it still would be interesting to know the typical ranges of the content of mast cells in bone marrow aspirates from healthy donors and systemic mastocytosis patients.
Answer to comment 3.- The section of the methods on the KIT mutation and FACS-sorting have been deleted in the new revised version of the manuscript, following the indication of the reviewer. Appropriate changes in the reference list have also been made. In addition, the distribution (i.e. range) of MC in nomal vs SM bone marrow samples have been added in material and methods section of the revised manuscript.
Comment 4.- The source of confusion in the definition of the stain index was the term “standard deviation”, which is ambiguous, since here it means signal spread, but perhaps more frequently it refers to inter-sample variation. Therefore I propose to use in the definition the up-dated terminology used by Maecker & Trotter, 2008, Nature Methods, saying something like “Stain index is the difference in signal width between the positive and the negative population divided by the signal spread of the negative population.” I think that this Nature Methods article, would be more appropriate than reference 61, since it efficiently explains the staining index by means of an illustration.
Answer to comment 4.- The definition provided by Maecker & Trotter, in Nature Methods 2008 is now used in table 1 and the text of the new revised version of the manuscript (materials and methods), as per the suggestion of the reviewer. This reference instead of the previous reference is now cited.
Comment 5.- Thanks for replacing the symbols in Fig. 2 by connecting lines. In my opinion, this shows the statistical differences between the populations in the box plot presentation in a more user-friendly manner. I do not think that it is necessary to repeat the stain index values and their p-values in a Table, particularly not in the main text.
Answer to comment 5.- The stain index values have been deleted in the table and the text of the new revised version of the manuscript, such stain index values being only shown now in the figure.
Additional minor remarks:
Remark 1.- It is more exact and therefore preferable to speak of “six analyzed surface proteins” rather than a “relatively large panel” (lines 27/p. 1 and 38/p. 3).
Answer to remark 1.- The sentence “six analyzed surface proteins” is now used in page1/line 27 and in page 3,line 38.
Remark 2.- The spelling of Gemtuzumab ozogamycin in Table 1 must be corrected.
Answer to remark 2.- The spelling of Gemtuzumab ozogamicin in Table 1 has been corrected.
Remark 3.- “Despite” in lines 38/p.11 and 32/p.12 must be replace by “although”.
Answer to remark 3.- The word “Despite” has been replaced by the word “although” in line 38/p.11. Line 32/p.12. has been deleted.
Remark 4.- The reference list could be shortened.
Answer to remark 4.- The reference list has been shortened, following the suggestion of the reviewer.